# Incidence and risk factors for medical care interruption in people living with HIV in a French provincial city

Anna Lucie Fournier[1,2]*, Jean-Jacques Parienti[1,2], Karen Champenois[3], Philippe Feret[2], Emmanuelle Papot[3], Yazdan Yazdanpanah[3,4], Renaud Verdon[1,2]

**1** GRAM 2.0, EA2656, UNICAEN, Normandie University Hospital, Caen, France, **2** Infectious Diseases Department, UNICAEN, Normandie University Hospital, Caen, France, **3** IAME, UMR1137, INSERM, Paris Diderot University, Sorbonne Paris Cité, Paris, France, **4** Infectious Diseases Department, AP-HP, Hôpital Bichat, Paris, France

* annafournier87@gmail.com

**Data Availability Statement:** All relevant data are within the paper.

**Funding:** The author(s) received no specific funding for this work.

## Abstract

### Objectives

The aim of our study was to identify HIV-positive patients at risk of medical care interruption (MCI) in a provincial city of a high-income country.

### Methods

We estimated the incidence rate of MCI in 989 individuals followed in an HIV clinic in Caen University Hospital, Normandy, France, between January 2010 and May 2016. We enrolled patients over 18 years old who were seen at the clinic at least twice after HIV diagnosis. Patients were considered to be in MCI if they did not attend care in or outside the clinic for at least 18 months, regardless of whether or not they came back after interruption. We investigated sociodemographic, clinical and immunovirological characteristics at HIV diagnosis and during follow-up through a Cox model analysis.

### Results

The incidence rate of MCI was estimated to be 3.0 per 100 persons-years (95% confidence interval [CI] = 2.6–3.5). The independent risk factors for MCI were a linkage to care >6 months after HIV diagnosis (hazard ratio [HR] = 1.14; 95% CI = 1.08–1.21), a hepatitis C coinfection (HR = 1.76; 95% CI = 1.07–2.88), being born in Sub-Saharan Africa (HR = 2.18; 95% CI = 1.42–3.34 vs. in France) and not having a mailing address reported in the file (HR = 1.73; 95% CI = 1.07–2.80). During follow-up, the risk of MCI decreased when the patient was older (HR = 0.28; 95% CI = 0.15–0.51 when >45 vs. ≤ 30 years old) and increased when the patient was not on antiretroviral therapy (HR = 2.78; 95% CI = 1.66–4.63).

### Conclusions

Our findings show that it is important to link HIV-positive individuals to care quickly after diagnosis and initiate antiretroviral therapy as soon as possible to retain them in care.

**Competing interests:** I have read the journal's policy and the authors of this manuscript have the following competing interests: JJP reports grants from ViiV Healthcare and MSD, and personal fees from Gilead, ViiV Healthcare and MSD outside the present work. KC has served as a speaker and as a consultant for Gilead outside the present work. YY has served as a speaker and as a consultant for Abbott, Bristol-Myers Squibb, Gilead, MSD, Roche, Tibotec and ViiV Healthcare outside the present work. RV received travel grants from Gilead, Merck and ViiVHealthCare outside the present work. ALF, PF and EP have no conflicts of interest to declare. This does not alter our adherence to PLOS ONE policies on sharing data and materials.

## Introduction

The UNAIDS goals for 2020 were that 90% of all people living with HIV (PLWH) know their HIV status, 90% of all people with diagnosed HIV infection receive sustained antiretroviral therapy (ART), and 90% of all people receiving ART have viral suppression [1]. Continuing engagement in care was the cornerstone of achieving 90% of viral suppression in treated PLWH by 2020 [2–4].

A regular medical follow-up is crucial to monitor the physical and mental condition of PLWH, to maintain ART adherence, and therefore to prevent HIV transmission [5–7]. Thus, improving retention in care of PLWH is a major public health issue with individual and societal outcomes.

Currently, ART is effective and enables CD4 levels to increase enough to achieve a life expectancy for PLWH similar to the general population [8, 9]. Now that HIV has become a chronic disease with lifelong treatment, the challenge is to maintain, enhance, and facilitate retention of HIV-positive patients in the healthcare system.

Medical care interruption (MCI) leads to immunity loss, HIV-related complications, and eventually death [10]. Previous studies have highlighted the risk factors at HIV diagnosis of definitive loss to follow-up [11, 12]. Only a few studies have assessed PLWH at risk of MCI (temporary or definitive) not only at HIV diagnosis but also during the follow-up [10, 13], including a study we performed in a large cohort of PLWH at Bichat university hospital in Paris, with 4,789 patients. We estimated an incidence rate of MCI of 2.5 per 100 persons-years and identified risk factors of MCI [14]. Some results were new, such as patients born in France were more at risk of MCI than those born in sub-Saharan Africa and not having reported a primary care physician found as a risk factor for MCI.

We then conducted the same methodology in a cohort of patients living in a 100,000-inhabitant provincial city in Normandy to estimate incidence rate and risk factors of MCI in a context other than Paris.

## Materials and methods

We used the same methodology as our previous cohort study [14]. Briefly, in a hospital cohort of patients followed for their HIV infection, we retrospectively searched for MCI in the medical file.

### Description of the cohort

Participants come from the HIV clinical cohort of the Infectious diseases department of the Caen university hospital, France. Caen is a 100,000-inhabitant city, located approximately 200 km west of Paris and is the head of southwestern Normandy. PLWH enrolled in this analysis were confirmed to be HIV infected from 1992 to 2014 by western blot, were ≥18 years old, attended the clinic at least twice between the 1st of January 2010 and the 31st of October 2014 and gave their informed consent for their medical data to be used in epidemiological analyses. Patients were followed through the 31st of May 2016.

### Definition of patients with medical care interruption

Patients were considered in MCI if they did not attend care in or outside the clinic for at least 18 months, regardless of whether or not they came back after interruption. We chose a longer time period than in other studies [12, 13, 15, 16] because French recommendations in 2013 advised one visit a year by an infectious diseases specialist [8]. Times between each contact with the clinic were calculated using the DMI2 computerized file completed by medical or

paramedical employees at each visit at the clinic. When the time between two contacts was over 18 months, the patient's medical record was reviewed to look for hospitalization, other medical contacts or death. For each remaining case of MCI, the patient and if no response, his primary care physician, was contacted by the infectious diseases referent to seek news, any medical contacts or blood tests. Thus, information on hospitalization in another department or another hospital, follow-up by another physician, relocation or death, during the period of no contact with the clinic were collected. If PLWH came back to care after an MCI with an HIV viral load (VL) under 50 copies/mL, we considered that they were still under ART and were not in MCI, assuming they were followed up elsewhere during this period [10].

## Variables

Data were collected prospectively through the DMI2 software for each patient by physicians and other healthcare workers. An electronic form was designed for the HIV follow-up and one file was created for each PLWH followed up at Caen hospital since 1992. For the analysis, we considered variables at HIV diagnosis and during the follow-up.

Clinical data at the time of HIV diagnosis were HIV transmission group (men who have sex with men-MSM, heterosexual, people who inject drug-PWID, or other), presence of Acquired Immune Deficiency Syndrome (AIDS) defined using the CDC classification, hepatitis C virus (HCV) and hepatitis B virus (HBV) coinfections and comorbidities (medical, surgical or psychiatric).

Variables recorded during the follow-up, at each patient visit, were clinical, biological and ART information. We modeled time-dependent age ($\leq$30 years, 31–45 years and >45 years), comorbidity, AIDS and coinfections status, CD4 cells and plasma VL, and ART prescription (Y/N).

Finally, we collected the referent physician's contact and the patient's last address if they were declared. The place of residency was used to reflect the distance from the hospital and the level of deprivation of the neighborhood as several studies found an association between MCI, geographical area and precarity [10, 17–19]. We classified people between four geographical areas centered around the clinic: Caen, the hospital city; Calvados, the hospital department; Normandy, the hospital district; outside Normandy; or without address available in the file.

All of the patients enrolled in this study gave their written informed consent to have their medical chart recorded in the electronic medical record system DMI2. The DMI2 database is funded by the French Ministry of Health and obtained a favorable opinion from the French Commission Nationale Informatique et Libertés (CNIL, authorization 1991/27/11) [20]. The ministerial decree authorizes the use of computerized medical and epidemiological records (of those patients who gave their informed consent) by the network of HIV medical centers caring for included patients for the areas of medical follow-up, epidemiological surveys and clinical research.

## Statistical analyses

We estimated the incidence rate of MCI as the number of PLWH who had one or more MCI, divided by the number of person-years at risk of MCI. Individuals were considered at risk of MCI from the date of enrolment in the study (1st January 2010 or date of their initial visit at the clinic if subsequent) to their last visit plus 6 months or not later than the 31st of May 2016, the end-point. A Poisson's exact distribution was used to estimate 95% confidence intervals (CIs).

We investigated sociodemographic, clinical and immunovirological characteristics at HIV diagnosis and during the time of follow-up through a Cox model to determine associations between MCI and variables.

The variables associated with MCI with a p-value <0.20 in the univariate analysis were entered into the multivariate model. The proportional hazard assumption was assessed using Shoenfeld residuals. We used a backward stepwise regression analysis to build the final multivariate model. Variables were considered significant if the p-value was <0.05 using Wald's test model. The interactions between variables that were significantly associated with MCI were studied two by two.

Statistical analyses were performed using STATA 14 (Stata Corporation, College Station, TX, USA).

## Results

Between the 1st of January 2010 and the 31st of October 2014, 989 individuals had at least two visits at the HIV department. Median duration of follow-up was 5.6 years [interquartile range (IQR), 2.97–6.08]. The sex ratio was 1.9 with 65% (646/989) men. The median age at enrollment was 45 years [IQR, 37–52]. The proportion of participants born in France was 81% (803/989), in sub-Saharan Africa was 13% (127/989) and in another country was 6% (59/989). A postal address was declared in 90% (894/989) of cases. A referring primary care physician was declared by 54% (539/989) of patients.

Demographic, clinical, biological and ART prescription characteristics at the time of HIV diagnosis are shown in Table 1. Median age was 33 years [IQR, 26–42] at HIV diagnosis. The median time between HIV diagnosis and the first medical contact for HIV was 1.6 months [IQR, 0.4–57.2] and under 6 months for 56% (557/989) of individuals. Half (50%, 493/988) of participants were heterosexual, 33% (324/988) MSM, and 13% (134/988) PWID. Eight percent (77/989) had AIDS at diagnosis, 4% (36/989) had an HBV infection and 10% (99/989) an HCV infection. The median of the first VL available was 11,672 copies/mL [IQR, 68–85,500]. The median of the first CD4 cell count available was 390 cells/mm3 [IQR, 202–554]. The median time between first medical visit for HIV and the first ART prescription was 28.1 months [IQR, 3.0–85.7].

### Incidence rate of MCI

Among the 989 patients included in the analysis, 284 patients interrupted their medical care for at least 18 months. For 205 patients, there was no visit recorded for 18 months and until the endpoint of the study (Fig 1). The infectious diseases specialist of these patients could be reached and then, the primary care physician for 77% of those declared by patients. Thus, 49 patients had died, 42 had moved to another region, and 5 were followed for HIV by a primary care physician. The infectious diseases specialists contributed to this information for 71% (68/96) and the primary care physicians for 29% (28/96). Finally, 11% (109/989) of patients had an MCI lasting at least until the end point.

In addition, 79 other patients had no visit recorded for ≥18 months during the follow-up but returned back into care during the study period. After reviewing medical records, 44 patients had an undetectable VL when they returned to care and were considered as followed elsewhere during this time period. Thus, 35 (4%) patients had at least one MCI and returned to care.

Overall, 144 of 989 (15%) patients were considered to have at least one MCI during the follow-up and among them, 35 (24%) returned to care during the study period. We estimated an

**Table 1. Characteristics of people infected with HIV seen at least twice at the infectious diseases department of Caen hospital between January 2010 and March 2014.**

| Characteristics | N = 989 |
|---|---|
| **Age at HIV diagnosis, years** | |
| Median [interquartile range] | 33 [26–42] |
| **Gender, n (%)** | |
| Male | 646 (65%) |
| Female | 343 (35%) |
| **Country of birth, n (%)** | |
| France | 803 (81%) |
| Sub-Saharan Africa | 127 (13%) |
| Other countries | 59 (6%) |
| **Geographical area[a], n (%)** | |
| Caen | 181 (18%) |
| Calvados | 362 (37%) |
| Normandie | 308 (31%) |
| Outside Normandie | 43 (4%) |
| No available address | 95 (10%) |
| **Primary care physician[b], n (%)** | |
| Yes | 539 (54%) |
| No | 450 (46%) |
| **Time period before first visit[c], n (%)** | |
| ≤ 6 months | 557 (56%) |
| > 6 months | 432 (44%) |
| **HIV transmission group, n (%)** | |
| Men who have sex with men | 324 (33%) |
| Heterosexual | 493 (50%) |
| People who inject drug | 134 (13%) |
| Other | 37 (4%) |
| **AIDS[d] at diagnosis, n (%)** | |
| Yes | 77 (8%) |
| No | 912 (92%) |
| **Hepatitis B coinfection, n (%)** | |
| Yes | 36 (4%) |
| No | 953 (96%) |
| **Hepatitis C coinfection, n (%)** | |
| Yes | 99 (10%) |
| No | 890 (90%) |
| **First available viral load (copies/mL), n (%)** | |
| <1,000 | 310 (31%) |
| 1,001–10,000 | 163 (17%) |
| 10,001–100,000 | 289 (29%) |
| >100,000 | 227 (23%) |
| **First available CD4 cells count (/mm$^3$), n (%)** | |
| ≤350 | 443 (45%) |
| >350 | 546 (55%) |
| **Time between first visit and first ART[e], n (%)** | |
| ≤12 months | 367 (37%) |

*(Continued)*

**Table 1.** (Continued)

| Characteristics | N = 989 |
|---|---|
| >12 months | 607 (61%) |

[a]Geographical area: from the address of residency declared by the patient and recorded in the electronic file.

[b]Primary care physician declared by the patient, and recorded in the electronic file, to whom the specialist letters are sent from the clinic.

[c]Time between HIV diagnosis and first medical visit in or outside the clinic.

[d]AIDS: Acquired Immune Deficiency Syndrome defined using the CDC classification.

[e]Time between HIV first care and first antiretroviral therapy (ART).

MCI incidence rate of 3.0 per 100 persons-years [95% CI = 2.6–3.5]. The median duration of follow-up in our study for people who interrupted care was 25.2 months [range, 0.2–73.6].

## Risk factors of MCI

The univariate analysis is shown in Table 2, and the multivariate analysis is shown in Table 3. No significant interactions between variables were found. Risk factors found to be independently associated with MCI were being born in sub-Saharan Africa (SSA; hazard ratio [HR] = 2.18; 95% CI = 1.42–3.34 vs. born in France), not having an available postal address (HR = 1.73; 95% CI = 1.07–2.80), a time before the first medical visit >6 months (HR = 1.14; 95% CI = 1.08–1.21) and an HCV coinfection (HR = 1.76; 95% CI = 1.07–2.88). During the follow-up, patients not under ART were more likely at risk of MCI (HR = 2.78; 95% CI = 1.66–

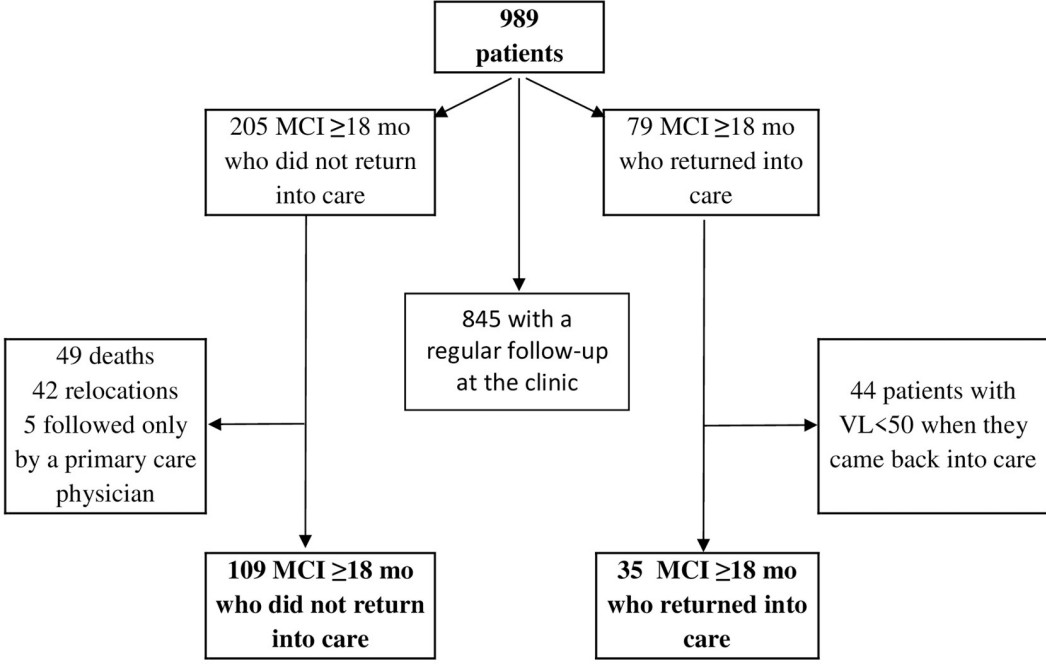

MCI: Medical Care Interruption

**Fig 1. Future of HIV-infected patients with no health care visit for at least 18 months.**

**Table 2. Univariate analysis of factors associated with MCI at time of HIV diagnosis and during follow-up in patients at Caen hospital followed between the 1st of January 2010 and the 31st of May 2016, n = 989 patients.**

| Factor | Hazard ratio (95% Confidence interval) | p-value |
|---|---|---|
| **Fixed variables** | | |
| **Age at HIV diagnosis** | 0.98 (0.96–0.99) | 0.01 |
| **Gender** | | |
| Male | 0.80 (0.57–1.11) | 0.19 |
| Female | 1 | - |
| **Country of birth** | | |
| France | 1 | - |
| Sub-Saharan Africa | 2.51 (1.70–3.71) | <0.0001 |
| Other countries | 1.10 (0.51–2.36) | 0.82 |
| **Available address, n (%)** | | |
| Yes | 1 | - |
| No | 2.30 (1.44–3.66) | 0.01 |
| **Geographical area[a]** | | |
| Caen | 1 | |
| Calvados | 0.75 (0.48–1.19) | 0.22 |
| Normandie | 0.73 (0.46–1.19) | 0.21 |
| Outside Normandie | 1.13 (0.52–2.46) | 0.76 |
| Missing data | 1.87 (1.07–3.27) | 0.03 |
| **Primary care physician[b]** | | |
| Yes | 1 | - |
| No | 1.32 (0.95–1.83) | 0.09 |
| **Time period before first visit[c]** | | - |
| ≤ 6 months | 1 | - |
| > 6 months | 1.09 (1.03–1.15) | 0.01 |
| **HIV transmission group** | | |
| Men who have sex with men | 1 | - |
| Heterosexual | 1.12 (0.77–1.64) | 0.54 |
| People who inject drug | 1.35 (0.81–2.24) | 0.24 |
| Other | 0.72 (0.26–2.01) | 0.53 |
| **Hepatitis B coinfection** | | |
| Yes | 1.13 (0.46–2.76) | 0.79 |
| No | 1 | - |
| **Hepatitis C coinfection** | | |
| Yes | 1.68 (1.06–2.67) | 0.04 |
| No | 1 | - |
| **First CD4 cells count (/mm3)** | | |
| ≤350 | 1 | - |
| >350 | 1.01 (0.99–1.01) | 0.64 |
| **Time before first ART[d]** | | |
| ≤12 months | 1 | - |
| >12 months | 1.17 (0.82–1.66) | 0.39 |
| **Time-dependent variables** | | |
| **Age (years)** | | |
| ≤30 | 1 | - |
| [31–45] | 0.64 (0.37–1.10) | 0.11 |
| >45 | 0.27 (0.15–0.47) | <0.0001 |

(*Continued*)

**Table 2.** (Continued)

| Factor | Hazard ratio (95% Confidence interval) | p-value |
|---|---|---|
| **ART** | | |
| Yes | 1 | - |
| No | 2.69 (1.65–4.39) | <0.0001 |
| **AIDS[e]** | | |
| Yes | 0.78 (0.51–1.21) | 0.27 |
| No | 1 | - |

[a]Geographical area: from the address of residency declared by the patient, and recorded in the computer file.

[b]Primary care physician declared by the patient, written in the computer file, whose consult letters are sent from the clinic.

[c]Time between HIV diagnosis and first medical visit in or outside the clinic.

[d]Time between HIV first care and first antiretroviral therapy (ART).

[e]AIDS: Acquired Immune Deficiency Syndrome defined by using the CDC classification.

**Table 3.** Multivariate analysis of factors associated with MCI at time of HIV diagnosis and during follow-up in patients at Caen Hospital followed between the 1[st] of January 2010 and the 31[st] of May 2016, n = 989 patients.

| Factor | Hazard ratio (95% Confidence interval) | p-value |
|---|---|---|
| **Fixed variables** | | |
| **Country of birth** | | |
| France | 1 | - |
| Sub-Saharan Africa | 2.18 (1.42–3.34) | <0.0001 |
| Other countries | 1.02 (0.47–2.21) | 0.97 |
| **Available address, n (%)** | | |
| Yes | 1 | |
| No | 1.73 (1.07–2.80) | 0.03 |
| **Hepatitis C coinfection, n (%)** | | |
| Yes | 1.76 (1.07–2.88) | 0.03 |
| No | 1 | - |
| **Time period before first visit[a]** | | |
| ≤ 6 months | 1 | - |
| > 6 months | 1.14 (1.08–1.21) | <0.0001 |
| **Time-dependent variables** | | |
| **Age (years)** | | |
| ≤ 30 | 1 | - |
| [31–45] | 0.62 (0.35–1.09) | 0.09 |
| > 45 | 0.28 (0.15–0.51) | <0.0001 |
| **ART[b]** | | |
| Yes | 1 | - |
| No | 2.78 (1.66–4.63) | <0.0001 |

No additional adjustment.

[a]Time between HIV diagnosis and first medical visit in or outside the clinic.

[b]Antiretroviral therapy prescription (ART).

4.63). Patients older than 30 years were less likely at risk of MCI (HR = 0.62; 95% CI = 0.35–1.09 when 31–45 vs. ≤ 30; and HR = 0.28; 95% CI = 0.15–0.51 when > 45 vs. ≤ 30).

## Discussion

Among the 989 PLWH followed in the HIV department in Caen, the incidence rate of having at least one MCI >18 months was estimated at 3.0 per 100 persons-years from 2010 to 2016. We found that 144 (15%) patients had an MCI, and among them, 109 (11%) were lost to follow-up until the end point of the study and 35 (24%) came back into care during the study period. Independent factors of increasing risk of MCI were being born in SSA, not having declared an available address, time before first medical visit >6 months after HIV diagnosis and HCV coinfection at HIV diagnosis. During follow-up, no ART was a risk factor for MCI and being older than 30 was a protective factor.

In France, few studies have been conducted to characterize PLWH with MCI. The incidence rate of MCI was estimated to be between 2.5 and 17.2 per 100 persons-years [10, 13–15, 21]. The highest incidence rate was in French Guiana where a lack in healthcare facilities, precarity and stigmatization can explain high MCI rates [21]. In the EuroSIDA cohort, the incidence rate of lost to follow-up (>12 months and no return to care) was estimated at 3.72 per 100 person-years, which was higher than in our cohort [12]. First, such a difference may be explained by the fact that the definition of MCI was stricter than ours (lost during 12 versus 18 months in our study and only medical visits count versus any contact with care in our study). Second, this study was conducted in different European countries, where care structures and precarity may be higher than in France. For example, Eastern Europe was an independent risk factor for MCI [HR = 1.78; 95% CI = 1.84–2.53].

In addition to the definition, the small number of patients in our clinic allowed for comprehensive care for PLWH, proactive methods to recall appointments and to bring back patients to care by phone calls or text messages that are currently used to secure retention in care. Indeed, patients who do not come to appointments are all actively sought by the infectious diseases specialist himself or a staff member (mail, phone calls).

The active search for patients initially thought to be lost to follow-up is one of the major strengths of our study. We could obtain information for 140 patients (49%) among the 284 ones considered first with an MCI thanks to the help of their infectious diseases specialist and primary care physician. We can underline the high level of answers among primary care physicians (77%) and the robustness of our methodology.

However, we found a higher incidence than in our previous research, which studied an HIV clinic cohort of 4,796 PLWH in Paris [HR = 2.5; 95% CI = 2.3–2.7] [14]. We wonder whether a larger center implements a stronger organization to monitor MCI in a timely manner or if it is an issue of accessibility. This difference could be explained by the distance to the hospital: the Caen clinic covers all of southwestern Normandy, with PLWH living up to 150 km away, compared to the Bichat HIV clinic in Paris, which covers a smaller but more densely populated area with more public transportation facilities. However, we did not find the geographical area of patients as a risk factor for MCI at the Caen clinic. Further studies should investigate whether geographical isolation plays a role in MCI.

Consistent with our findings, several authors have found that people from SSA countries were more likely to be at risk of MCI [13, 15, 16, 21–23]. The authors explained that immigrants are geographically mobile and found a higher proportion of people from sub-Saharan Africa among people with MCI who did not return to care. In another French cohort, Ndiaye et al. [13] found an adjusted HR of 1.81 (95% CI = 1.16–2.80) for these patients to be in MCI. In France, migrants can obtain state medical assistance (AME) and benefit from the

permanence of access to healthcare (PASS) since 1998 (Art. L. 711-7-1 of the Public Health Code) [24]. Even with a healthcare system providing free care for undocumented citizens, this group of PLWH remains more likely at risk of MCI. Interestingly, we found an opposite result in the Paris cohort with a lower risk of MCI among patients born in SSA or in a country other than France (HR = 0.75, 95% CI = 0.62–0.91 and HR = 0.79, 95% CI = 0.63–0.99, respectively) [14]. A higher proportion of people born outside France was followed in this center, which always had a large community of migrants from SSA (40% in the Parisian cohort versus 13% in this provincial one). Moreover, people born in SSA might find more help from the community in Paris and a larger social network, which may not be the case in less densely populated areas, such as Normandy. Further studies are needed to understand this factor with a social approach.

No postal address declared to the clinic was associated with a significantly higher risk of MCI. It is questionable whether patients without an informed address (10%) are homeless or at least housing unstable. Homeless people had a higher risk of MCI among 1,756 PLWH followed in Tourcoing (HR = 2.2, 95% CI = 1.0–4.9) [15].

HCV coinfection at the time of HIV diagnosis was found to be a risk factor. This result was also found by Giordano et al. [23] in a cohort study of 2,619 male US veterans regardless of PWID status.

Being over 30 during follow-up was a protective factor for MCI, and we found a trend for an even lower risk for patients over 45 years old. Young age is indeed often found in the literature as a major risk factor for MCI [10, 13, 15, 25]. The lifestyle of patients $\leq$ 30 years of age, conflictual relationships related to adolescence, and higher stigmatization of the sick individuals are barriers to regular medical follow-up [26]. Improving the retention of the youngest PLWH is an important public health issue, at the individual level by the number of years of life gained and at the collective level by the risk of HIV transmission induced by the sexual activity of people $\leq$ 30 years [27, 28].

Other reasons for MCI listed in the literature include lack of time, fear of stigma, childcare, transportation problems, forgetting the appointment, diseases or other family member's diseases [26]. Multiple visits at the beginning of treatment, before starting treatment, the costs and time of transport, employment restrictions, and family are structural constraints. Our results showed key populations, people born abroad, young individuals, without a postal address available and/or HCV coinfected, may suffer from these structural constraints but also from a lack of understanding of the disease and care and the need for monitoring.

The same risk factors found to be independently associated with an MCI in our two studies, in a capital or a provincial city, were delayed care initiation after HIV diagnosis and no ART prescribed during follow-up [14]. In the present study, the majority of the patients (54%) initiated ART more than one year after HIV diagnosis. They were diagnosed from 1992 for the oldest patients and included in our study before 2013 guidelines for the majority of our cohort. Indeed, since 2013, French and WHO guidelines are in favor of early antiretroviral initiation, regardless of CD4 count, which suggests that we could expect an improvement in retention in care [1, 8]. The treatment may act as a link between the healthcare system and the PLWH. In the UK, PLWH not on ART had a 5-fold higher risk of interrupting care than those on ART [29]. In addition to benefits of an early linkage to care and ART prescription on patient follow-up, several trials showed improvement in retention in care for a same day treatment, with ART given the day of HIV diagnosis [30–32]. In Haiti, 80% of patients who initiated ART on the day of diagnosis versus 71% of those who did not were still alive and in care at twelve months, and 54% versus 42% with an undetectable VL (p = 0.004) [30]. In South Africa, 81% of patients who initiated ART in the rapid arm versus 64% of patients in the standard arm

were retained in care at ten months [31]. In San Francisco, for 4/39 (10%) patients with MCI in the rapid arm versus 7/47 (15%), no significant difference was found [32].

Our study is limited by its retrospective design that may introduce bias in the classification of MCI and other factors due to missing data. Nevertheless, data were collected prospectively in the DMI2® database. We actively searched for patients with MCI, which reduced misclassification bias. However, we did not have any information on 13 patients over 144 considered with MCI. We may have overestimated the incidence rate of MCI by considering that dead people have interrupted care.

## Conclusions

Accelerating linkage to care and increasing retention of these key populations in the HIV care continuum is essential to reach the 95-95-95 UNAIDS goals by 2030 to end the epidemic [33]. Interventions to improve retention of PLWH in care should focus on patients identified at risk for MCI as in our study because there are regional differences across cohorts. More studies are needed to identify these differences and barriers PLWH at risk of MCIs struggle with to stay in care. These barriers are structural, behavioral and medical. Reducing the number of visits at the beginning of treatment and the time period before ART initiation are medical hardships. We must consider these issues to tailor optimal interventions.

## Acknowledgments

The authors warmly thank the Caen Hospital Department of Medical Information and the COREVIH Normandy for their logistical support. We also thank all infectious disease and primary care physicians involved in the study.

## Author Contributions

**Conceptualization:** Anna Lucie Fournier, Jean-Jacques Parienti, Yazdan Yazdanpanah, Renaud Verdon.

**Data curation:** Anna Lucie Fournier, Philippe Feret, Renaud Verdon.

**Formal analysis:** Anna Lucie Fournier.

**Funding acquisition:** Anna Lucie Fournier.

**Investigation:** Anna Lucie Fournier, Philippe Feret.

**Methodology:** Anna Lucie Fournier, Jean-Jacques Parienti, Yazdan Yazdanpanah.

**Resources:** Karen Champenois.

**Software:** Philippe Feret.

**Supervision:** Karen Champenois, Emmanuelle Papot.

**Validation:** Jean-Jacques Parienti, Renaud Verdon.

**Visualization:** Jean-Jacques Parienti, Yazdan Yazdanpanah.

**Writing – original draft:** Anna Lucie Fournier.

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
