## [Decision Letter · Decision Letter 0]

2 Sep 2020

PONE-D-20-20346

Incidence and risk factors for medical care interruption in people living with HIV in a French provincial city

PLOS ONE

Dear Dr. Fournier,

Thank you for submitting your manuscript to PLOS ONE. After careful consideration, we feel that it has merit but does not fully meet PLOS ONE’s publication criteria as it currently stands. Therefore, we invite you to submit a revised version of the manuscript that addresses the points raised during the review process.

We look forward to receiving your revised manuscript.

Kind regards,

Justyna Dominika Kowalska

Academic Editor

PLOS ONE

Journal Requirements:

2. Thank you for including your competing interests statement; "I have read the journal's policy and the authors of this manuscript have the following competing interests: JJP reports grants from ViiV Healthcare and MSD, and personal fees from Gilead, ViiV Healthcare and MSD outside the present work. KC has served as a speaker and as a consultant for Gilead outside the present work. YY has served as a speaker and as a consultant for Abbott, Bristol-Myers Squibb, Gilead, MSD, Roche, Tibotec and ViiV Healthcare outside the present work. RV received travel grants from Gilead, Merck and ViiVHealthCare outside the present work. ALF, PF and EP have no conflicts of interest to declare."

Additional Editor Comments (if provided):

Dear Dr Fournier, please provide text editing in English as suggested by the reviewer.

Reviewers' comments:

Reviewer's Responses to Questions

**Comments to the Author**

1. Is the manuscript technically sound, and do the data support the conclusions?

Reviewer #1: Yes

2. Has the statistical analysis been performed appropriately and rigorously? 

Reviewer #1: Yes

3. Have the authors made all data underlying the findings in their manuscript fully available?

Reviewer #1: Yes

4. Is the manuscript presented in an intelligible fashion and written in standard English?

Reviewer #1: Yes

5. Review Comments to the Author

Reviewer #1: The study titled "Incidence and risk factors for medical care interruption in people living with HIV in a French provincial city" PONE-D-20-20346 is a valuable work. Continuity of medical care is very important, especially in PLWH because of regular antiretroviral treatment and immunodeficiency - they are special patients. That is why the topic about medical care interruption among PLWH is very interesting.

Abstract contains the most important issues. The introduction is thoughtful and leads to the main topic. The research methods are clear and the patients cohort is quite large. Statistical analyses are written well. The results are detailed and the tables are in proper format. The discussion The limitations of the study are included as well. The discussion compares studies not only from France but also from other countries, which is additional value of this study.

I would suggest to check the paper by native speaker - there are some small English language mistakes

6. PLOS authors have the option to publish the peer review history of their article (what does this mean?). If published, this will include your full peer review and any attached files.

Reviewer #1: No

---

## [Author Response · Author response to Decision Letter 0]

24 Sep 2020

We thank the editor for giving us the opportunity to present our study, as well as the reviewer for his(her) comments. 

Journal Requirements:

The revised manuscript was edited to meet PLOS ONE’s style requirements.

Here is our updated Competing Interests statement as required:

“I have read the journal's policy and the authors of this manuscript have the following competing interests: JJP reports grants from ViiV Healthcare and MSD, and personal fees from Gilead, ViiV Healthcare and MSD outside the present work. KC has served as a speaker and as a consultant for Gilead outside the present work. YY has served as a speaker and as a consultant for Abbott, Bristol-Myers Squibb, Gilead, MSD, Roche, Tibotec and ViiV Healthcare outside the present work. RV received travel grants from Gilead, Merck and ViiVHealthCare outside the present work. ALF, PF and EP have no conflicts of interest to declare. This does not alter our adherence to PLOS ONE policies on sharing data and materials.”

Editor's comment:

As the editor required, the revised manuscript was edited for proper English language, grammar, punctuation, spelling, and overall style by one or more of the highly qualified native English speaking editors at American Journal Experts.

Reviewer' comment:

A native-English speaking editors at American Journal Experts reviewed this version of the manuscript. A certificate issued on September 18, 2020 may be verified on the AJE website using the verification code 2E8D-FB26-C6A6-1AE2-8F86.

---

## [Editor Report · Decision Letter 1]

28 Sep 2020

Incidence and risk factors for medical care interruption in people living with HIV in a French provincial city

PONE-D-20-20346R1

Dear Dr. Fournier,

We’re pleased to inform you that your manuscript has been judged scientifically suitable for publication and will be formally accepted for publication once it meets all outstanding technical requirements.

Kind regards,

Justyna Dominika Kowalska

Academic Editor

PLOS ONE
---

## [Editor Report · Acceptance letter]

5 Oct 2020

PONE-D-20-20346R1 

Incidence and risk factors for medical care interruption in people living with HIV in a French provincial city 

Dear Dr. Fournier:

I'm pleased to inform you that your manuscript has been deemed suitable for publication in PLOS ONE. Congratulations! Your manuscript is now with our production department. 

Kind regards, 

on behalf of

Dr. Justyna Dominika Kowalska 

Academic Editor

PLOS ONE